# AGO-bound mature miRNAs are oligouridylated by TUTs and subsequently degraded by DIS3L2

Acong Yang [1,4], Tie-Juan Shao[1,2,4], Xavier Bofill-De Ros [1,4], Chuanjiang Lian[1,3], Patricia Villanueva[1], Lisheng Dai[1] & Shuo Gu [1✉]

MicroRNAs (miRNAs) associated with Argonaute proteins (AGOs) regulate gene expression in mammals. miRNA 3′ ends are subject to frequent sequence modifications, which have been proposed to affect miRNA stability. However, the underlying mechanism is not well understood. Here, by genetic and biochemical studies as well as deep sequencing analyses, we find that AGO mutations disrupting miRNA 3′ binding are sufficient to trigger extensive miRNA 3′ modifications in HEK293T cells and in cancer patients. Comparing these modifications in TUT4, TUT7 and DIS3L2 knockout cells, we find that TUT7 is more robust than TUT4 in oligouridylating mature miRNAs, which in turn leads to their degradation by the DIS3L2 exonuclease. Our findings indicate a decay machinery removing AGO-associated miRNAs with an exposed 3′ end. A set of endogenous miRNAs including miR-7, miR-222 and miR-769 are targeted by this machinery presumably due to target-directed miRNA degradation.

[1] RNA Mediated Gene Regulation Section; RNA Biology Laboratory, Center for Cancer Research, National Cancer Institute, Frederick, MD 21702, USA. [2] School of Basic Medicine, Zhejiang Chinese Medical University, Hangzhou 310053, China. [3] State Key Laboratory of Veterinary Biotechnology and Heilongjiang Province Key Laboratory for Laboratory Animal and Comparative Medicine, Harbin Veterinary Research Institute, Chinese Academy of Agricultural Sciences, Harbin 150069, China. [4] These authors contributed equally: Acong Yang, Tie-Juan Shao, Xavier Bofill-De Ros. ✉email: shuo.gu@nih.gov

MicroRNAs (miRNAs) are a class of small non-coding RNAs that serve as master regulators of gene expression in eukaryotic cells[1]. miRNAs bind to Argonaute proteins (AGO), with their 5′ end embedded inside the AGO MID domain and the 3′ end docked at the AGO PAZ domain[2]. The AGO-miRNA complex forms the core of the RNA-induced silencing complex (RISC). Through partial base-pairing, miRNAs guide the RISC to target mRNAs, downregulating their levels by translational repression and/or mRNA degradation[3]. Most human mRNAs contain at least one functional miRNA target site, indicating that nearly all biological pathways are under miRNA regulation[4]. Not surprisingly, dysregulation of miRNAs is associated with and potentially leads to human diseases[5].

While layers of regulation in miRNA biogenesis have been characterized[6], little is known about miRNA turnover[7]. Due to Argonaute protection, miRNAs are generally stable in cells, with half-lives ranging from hours to days[8–11]. Therefore, active decay is critical in situations that require rapid changes in miRNA function[12]. Tudor-SN cleaves AGO-bound miRNAs containing CA and UA dinucleotides, playing an important role in regulating cell cycle transition[13]. Another mechanism to achieve miRNA-specific decay is target-directed miRNA degradation (TDMD), in which extensively paired targets induce miRNA turnover[14]. TDMD was initially described to be triggered by artificial targets as well as viral RNAs[15,16]. Recently, endogenous transcripts such as lncRNA CYRANO and NREP mRNA were shown to downregulate miR-7 (ref. [17]) and miR-29 (ref. [18]), respectively, via TDMD, indicating that TDMD could be a general mechanism regulating miRNA stability.

miRNAs are thought to be degraded from the 3′ end. During TDMD, extensive pairing between miRNA and targets promotes the dislocation of the 3′ end of miRNA from AGO binding and makes it accessible to enzymatic modifications[19]. As a result, TDMD-induced miRNA turnover is often accompanied by elevated levels of miRNA 3′ isoforms (3′ isomiRs)[14,15]. A set of terminal nucleotidyltransferases (TENTs), including TUT4 (ZCCHC11/TENT3A/PAPD3) and TUT7 (ZCCHC6/TENT3B/PAPD6) are responsible for adding non-templated nucleotides to the 3′ ends of miRNAs (tailing), whereas 3′–5′ exonucleases shorten the miRNAs by removing nucleotides from the 3′ end (trimming)[20–22]. 3′ uridylated or adenylated isomiRs have different stabilities compared to the cognate miRNAs[23–25] and plant miRNAs uridylated by HESO1 and URT1 at the 3′ termini are removed by 3′–5′ exonucleases[26,27]. However, it remains unclear how the stability of miRNAs is regulated by 3′ modifications in animals.

Here, we investigate how 3′ modifications lead to miRNA decay by generating mutations in the AGO PAZ domain. These mutations are sufficient to trigger extensive miRNA 3′ end modifications in cultured cells and in vivo, suggesting a machinery monitoring the status of the miRNA 3′ end. We find that these AGO-bound miRNAs with exposed 3′ ends are oligouridylated by both TUT4 and TUT7 and subsequently degraded by DIS3L2. Interestingly, abolishing oligo-tailing resulted in elevated trimming, suggesting that a tailing-independent trimming process functions redundantly in removing AGO-bound miRNAs with an exposed 3′ end. We provide evidence that a set of endogenous miRNAs, including miR-7, miR-222, and miR-769, which are likely under regulation of TDMD, are targeted by the TUT-DIS3L2 machinery in HEK293T cells.

## Results

**Disrupting the AGO PAZ domain promotes miRNA 3′ modification**. We sought to test whether dislocating the miRNA 3′ end from AGO is sufficient to promote 3′ modifications. To this end, we created an AGO2 mutant (AGO2-F2L3) containing two point mutations at the core of the PAZ domain binding pocket: F294A and L339A[28]. We co-expressed FLAG-tagged AGO2-F2L3 in HEK293T cells with miR-27a, a miRNA known to be regulated by TDMD during viral infection[16]. Mature miR-27a associated with AGO2 mutant was pulled-down via immunoprecipitation and detected by Northern blot. Results were AGO-specific since we detected no signal in the GFP pull-down control. We observed reduced miR-27a levels as well as extensive isomiRs with different electrophoretic mobilities in the pull-downs of the PAZ mutant (AGO2-F2L3) but not in those of wild-type AGO2 (AGO2-WT) or a version of AGO2 mutated at the catalytic center (AGO2-D597A) (Fig. 1a). Deep sequencing analyses confirmed that the aberrantly sized RNAs consisted of miR-27a of variable lengths (Supplementary Fig. 1a). Mapping reads to the genome sequence revealed that those miR-27a isomiRs were a result of 3′ modifications, consisting of 3′ trimming, tailing or both (Fig. 1b, Supplementary Fig. 1b). Consistent with previous studies of miRNA 3′ modifications[19,29,30], non-templated nucleotides added to the 3′ end of miR-27a were mainly U or A (Fig. 1b, Supplementary Fig. 1b).

To rule out the possibility that these 3′ modifications occurred on pre-miRNAs and were inherited by mature miRNAs, we repeated the same experiment with a synthetic miR-27a duplex, which is loaded to AGO2 directly (Supplementary Fig. 1c). We observed extensive 3′ tailing and trimming with the guide strand miR-27a but not with the passenger strand (Supplementary Fig. 1d). Because the former but not the latter is retained in mature RISC (Supplementary Fig. 1c), this result indicates that the 3′ modifications occurred on the mature miR-27a. Co-expressing AGO2-F2L3 with two additional 3p miRNAs (miR-23a and miR-1) and two additional 5p miRNAs (miR-122 and miR-15) generated similar results (Fig. 1c). Since the 3′ end of 5p miRNAs is unavailable in the pre-miRNA (Supplementary Fig. 1c), these results confirmed that the AGO PAZ mutation promotes modifications on AGO-bound mature miRNAs. Consistent with a recent report[19], these results indicate that releasing the 3′ end from the PAZ domain is sufficient to induce 3′ modifications. AGO PAZ mutants thus provide a unique platform for studying the 3′ modifications of miRNAs.

**Endogenous miRNAs with exposed 3′ ends undergo modification**. Next, we deep-sequenced endogenous miRNAs in the pull-downs of either AGO2-WT or AGO2-F2L3. We observed increased 3′ modification of miRNAs associated with AGO2-F2L3 compared to those associated with AGO2-WT (Fig. 2a), suggesting that the PAZ domain protects the 3′ terminus of AGO-bound miRNAs from being modified. Given that this phenomenon occurs for nearly all miRNAs, the underlying machinery is unlikely to be sequence-specific. 5p miRNAs and 3p miRNAs had a similar degree of 3′ modification (Fig. 2b), indicating that the majority of these modifications occur at the level of mature miRNA. These 3′ modifications mainly consisted of trimming (Fig. 2c, Supplementary Fig. 2a). As a result, shorter isomiRs were enriched in the pull-downs of the AGO2 PAZ mutant (Fig. 2d). We validated these observations by Northern blot analyses of three abundant endogenous miRNAs (miR-10b, miR-7, and miR-148a) (Fig. 2e). Compared to the results obtained with overexpressed miRNAs (Fig. 1), the apparent lack of 3′ tailing suggested that tailed isomiRs of endogenous miRNAs are efficiently removed in cells.

To extend our conclusion beyond cultured cells and artificially introduced AGO2 PAZ mutations, we took advantage of the datasets deposited in The Cancer Genome Atlas (TCGA) where various mutations in AGO are documented and the

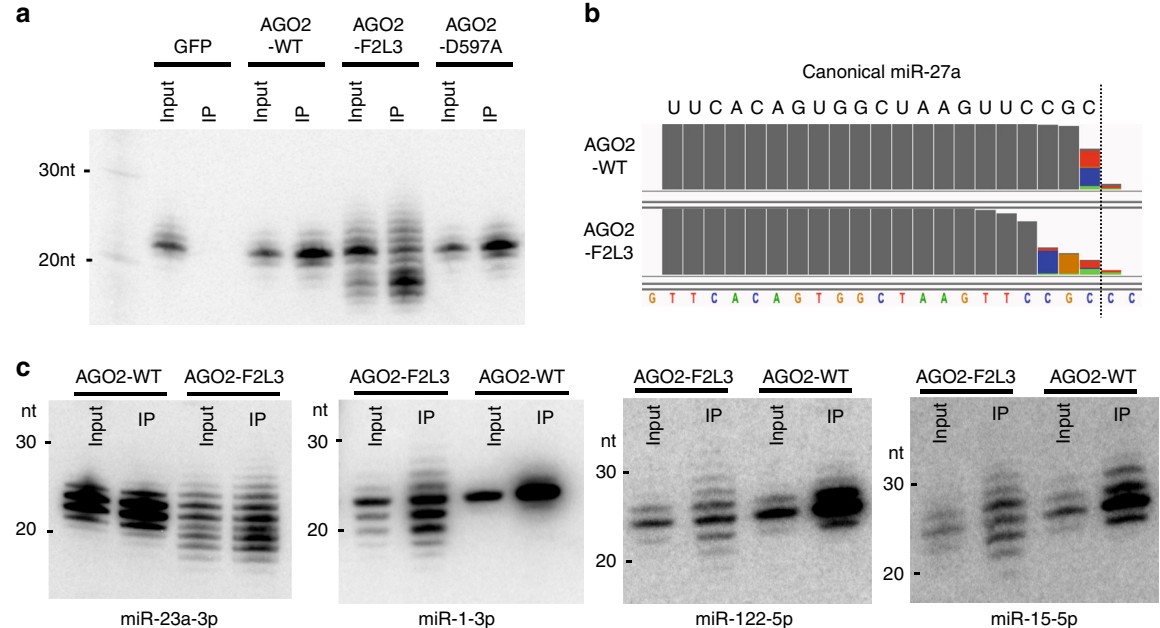

**Fig. 1 Disrupting the AGO PAZ domain promotes miRNA 3′ modification. a** Co-expression of pri-miR-27a and FLAG-AGO2 constructs: wild-type (WT), PAZ mutant (F2L3), and slicer-dead (D597A). Detection of miR-27a-3p in input and FLAG-immunoprecipitate (IP) by Northern blot. **b** Sequence composition of miR-27a-3p reads bound to AGO2-WT and AGO2-F2L3 on IGV (integrative genomics viewer). **c** Co-expression of FLAG-AGO2 constructs and miRNAs with preferential loading of 3p and 5p arms (miR-23a-3p, miR-1-3p, miR-122-5p, and miR-15-5p). Detection of mature miRNA in input and FLAG-immunoprecipitate (IP) by Northern blot. Source data are provided as a Source Data file.

corresponding miRNA sequencing data are available. miRNA 3′ modifications were more prevalent in a patient bearing an AGO2 mutation at the PAZ 3′ binding pocket (P295L) than that in patients with AGO2 synonymous or missense mutations at other regions of AGO2 (Fig. 2f, Supplementary Fig. 2b). A similar result was obtained in a patient with another AGO2 mutation in the PAZ 3′ binding pocket (R315M) (Supplementary Fig. 2b, c), but not in a patient with a mutation outside (E299K) (Supplementary Fig. 2b, d), indicating that the increased 3′ modifications are specific to the miRNA 3′ end dislocation. A mutation at the 3′ binding pocket of AGO1 (F310L) led to a similar result (Supplementary Fig. 2e, f), demonstrating that 3′ modifications are not limited to AGO2-bound miRNAs. Taken together, our results suggest a cellular machinery surveilling the 3′ end of AGO-bound miRNAs. Once exposed, 3′ ends of miRNAs are subjected to extensive sequence modifications.

**TUT4 and TUT7 oligouridylate the 3′ end of mature miRNAs.** By taking advantage of the AGO PAZ mutant, we sought to identify the enzymes responsible for tailing mature miRNAs. To facilitate detection of the tailed products, we generated a truncated form of AGO2 with a complete PAZ domain deletion (AGO2-ΔPAZ). We reasoned that AGO2-ΔPAZ would make the 3′ end of miRNAs fully exposed and therefore increase the robustness of tailing. Indeed, we observed extensive tailing by Northern blot when miR-27a was co-expressed with AGO2-ΔPAZ. We detected a ladder of bands with sizes of around 25–40 nt using a probe against mature miR-27a sequence (Fig. 3a), but not with probes against the miR-27a passenger strand or loop sequence (Supplementary Fig. 3a), confirming that these bands were mature miR-27a with long tails.

To further characterize these tailed products, we deep sequenced small RNAs in the pull-downs of AGO2-ΔPAZ. Despite exhaustive efforts, we failed to clone the long-tailed isomiRs using conventional methods. We therefore increased the

incubation time of 3′ adapter ligation during library preparation (see "Methods" for details). This procedure, which enabled the detection of isomiRs with long-tails, also led to nonspecific ligations between miR-27a and various RNA fragments in cells, resulting in a high background of nonspecific tail sequences. To distinguish tailed isomiRs from noise, we compared the sequencing results of AGO2-ΔPAZ-IP to those of AGO2-WT-IP, where long-tailed isomiRs were absent in the Northern blots (Fig. 3a). We reasoned that tail sequences detected in the AGO2-WT-IP represent the background whereas sequences enriched in the AGO2-ΔPAZ-IP are derived from authentic miR-27a tails. Using this method, we found that U and, to a much lesser extent, A, but not G or C, were enriched in the miR-27a oligo-tails (Fig. 3b, Supplementary Fig. 3b). The same experiment and analysis performed with miR-134-5p generated a similar result (Supplementary Fig. 3c), indicating that long-tail isomiRs are largely a result of oligouridylation.

TUT4 and TUT7 oligouridylate a set of pre-miRNAs to regulate their biogenesis[31–33]. We and others also showed that TUT4 and TUT7 can uridylate mature miRNAs in vitro and in vivo[34,35]. Taking advantage of a previously established TUT4 and TUT7 double knockout (TUT4/7 DKO) HEK293T cell line[35], we tested whether TUT4 and TUT7 are the enzymes responsible for the robust oligouridylation of AGO-bound miRNAs with an exposed 3′ end. The accumulation of long-tailed miR-27a isomiRs resulting from co-expression of AGO2-ΔPAZ was abolished in the TUT4/7 DKO cells (Fig. 3c "GFP lane"). This was rescued by ectopic expression of either TUT4 or TUT7 (Fig. 3c). Oligo-tailing of miR-27a resulting from the co-expression of AGO2-F2L3, although less robust than that resulting from co-expression of AGO2-ΔPAZ, was impacted by TUT4 and TUT7 in a similar manner (Fig. 3d). The same experiment performed with miR-23a generated a similar result (Supplementary Fig. 3d). Together, these results indicate that TUT4 and TUT7 are the major enzymes oligouridylating AGO-bound miRNAs once their 3′ ends are exposed.

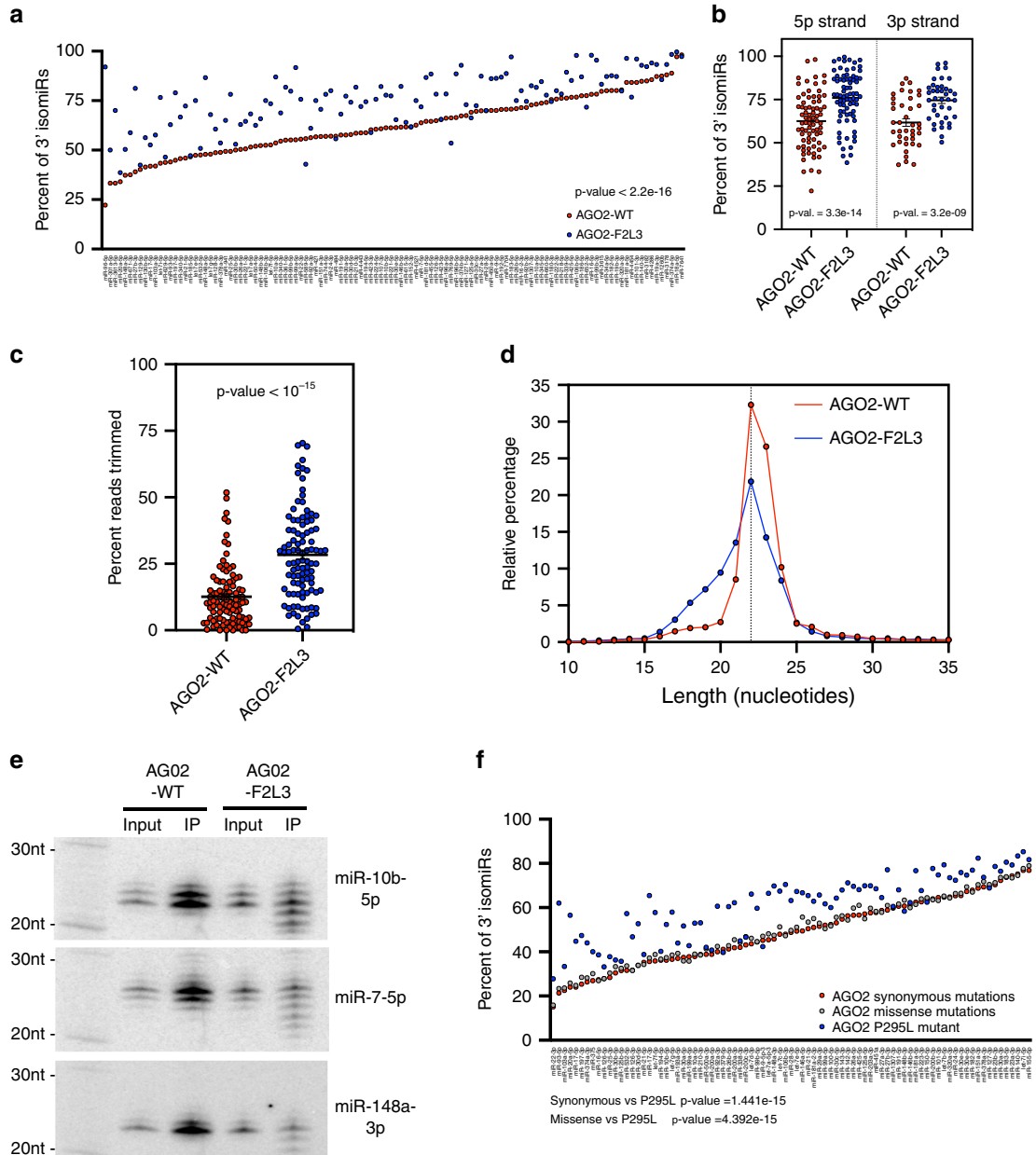

**Fig. 2 Endogenous miRNAs with exposed 3′ ends undergo modification. a** Plot of percentage of isomiRs (*y*-axis) for the endogenous miRNA with highest expression (*x*-axis, *N* = 116 miRNAs) bound to AGO2-WT and AGO2-F2L3. Wilcoxon's test for paired values was used to calculate the *p* value. **b** Plot of percentage of isomiRs. Unpaired Student's *t* test was used for comparing endogenous miRNAs associated with AGO2-WT from miRNAs associated with AGO2-F2L3 (mean ± SEM, $N_{5p\text{-arm}}$ = 76 miRNAs, $N_{3p\text{-arm}}$ = 40 miRNAs). **c** Plot of percentage of trimmed reads in endogenous miRNA (mean ± SEM, *N* = 100 miRNAs). **d** Histogram of the length distribution in endogenous miRNA bound to AGO2-WT and AGO2-F2L3. **e** Detection of endogenous mature miRNA in input and FLAG-immunoprecipitate (IP) by Northern blot. **f** Plot of percentage of isomiRs for the endogenous miRNA with highest expression (*N* = 85) in TCGA samples with synonymous (51 patients), missense (81 patients), and P295L mutation on AGO2. Wilcoxon's test for paired values (two-sided) were used to calculate the *p* values. Source data are provided as a Source Data file.

**TUT7 associates with Argonaute proteins.** Although expressing either TUT4 or TUT7 rescued oligouridylation of miRNAs in TUT4/7 DKO cells, TUT7-mediated tails were substantially longer (Fig. 3c, d, Supplementary Fig. 3d), suggesting that TUT7 is more robust. To test this, we knocked out TUT4 and TUT7 individually in HEK293T cells, confirming loss of their expression by Western blot (Supplementary Fig. 4a). While knocking out TUT7 diminished the long tails of miR-27a associated with the AGO2-ΔPAZ, depletion of TUT4 had marginal impact on the oligouridylation of miR-27a (Fig. 4a). We observed a similar pattern with AGO2-F2L3-induced tails (Supplementary Fig. 4b)

as well as oligouridylation of miR-23a (Supplementary Fig. 4c), demonstrating that TUT7 is more robust than TUT4 in oligouridylating miRNAs with an exposed 3′ end.

TUT4 and TUT7 have a similar potency in uridylation of naked miRNA in vitro[34]. It is also known that their oligouridylation activity is increased when the enzymes are anchored to the substrate pre-let-7 via a bridge protein LIN28[36,37]. We therefore hypothesized that the observed superior oligouridylation activity of TUT7 is due to its stronger association with the RISC. Indeed, we detected endogenous TUT7 but not TUT4 in the immunoprecipitation pull-downs of FLAG-tagged AGO2-WT,

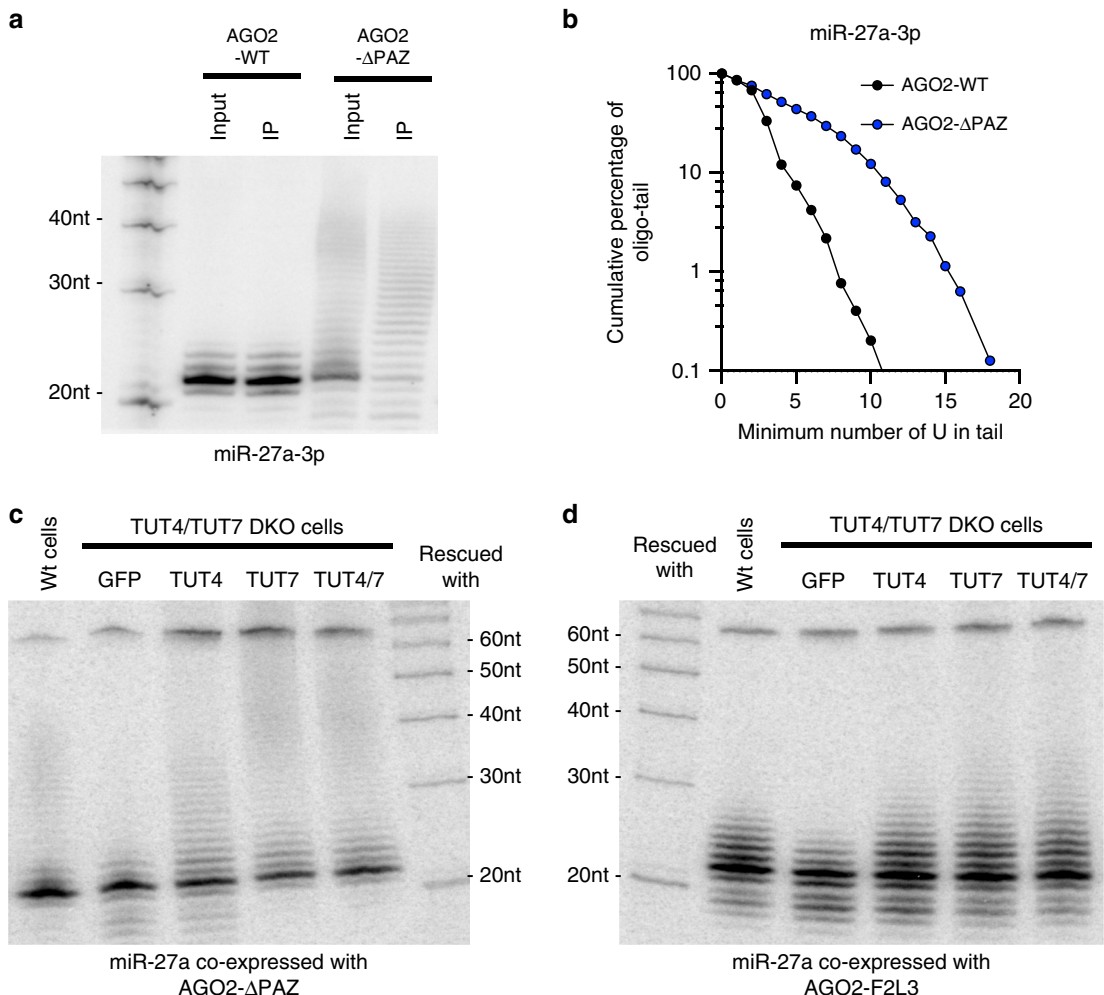

**Fig. 3 TUT4 and TUT7 oligouridylate the 3′ end of mature miRNAs. a** Co-expression of pri-miR-27a and FLAG-AGO2 constructs: wild-type (WT) and PAZ domain deletion (ΔPAZ). Detection of miR-27a-3p in input and FLAG-immunoprecipitate (IP) by Northern blot. **b** Nucleotide composition of miR-27a-3p oligo-tail. Horizontal axis indicates the absolute number of non-templated U nucleotides in tail, and the vertical axis indicates the cumulative percentage from the longest to the shortest oligo-tails. **c, d** Co-expression of pri-miR-27a and AGO2-ΔPAZ (**c**) or AGO2-F2L3 (**d**) in wild-type HEK293T or TUT4/7 DKO, and TUT4/7 DKO cells rescued with either GFP, TUT4, TUT7, or TUT4 and TUT7. Detection of miR-27a-3p by Northern blot. Source data are provided as a Source Data file.

AGO2-F2L3, and AGO2-ΔPAZ, but not in the pull-downs of FLAG-GFP with or without RNase treatment (Fig. 4b), indicating that TUT7 interacts with AGO2 in an RNA-independent manner. To confirm that the observed interaction was not due to overexpression of AGO2, we performed the reverse co-immunoprecipitation assay in HEK392T cells without ectopic expression of either AGO or TUT. Supporting our conclusion, AGO1 and AGO2 were detected in the pull-downs of TUT7 but not in those of TUT4 (Fig. 4c). However, given the limitations in the sensitivity of our assay, we cannot rule out the possibility that TUT4 also interacts with AGOs in cells. Only a small fraction of TUT7 and AGO2 were in complex with each other (Fig. 4b, c), which is expected since TUT7 targets many other RNAs[22] and only those AGO2-associated miRNAs with accessible 3′ ends are TUT7 targets.

**DIS3L2 degrades oligouridylated mature miRNAs**. DIS3L2 removes pre-miRNAs oligouridylated by TUT4 and TUT7[38]. To test whether DIS3L2 plays a similar role in degrading oligouridylated mature miRNAs, we generated a HEK293T DIS3L2 KO cell line using CRISPR–Cas9 and confirmed loss of DIS3L2 expression by Western blot (Supplementary Fig. 5a). Upon co-

expression of miR-27a with AGO2-ΔPAZ in these cells, oligouridylated forms of miR-27a accumulated that were increased in length and intensity compared to WT cells (Fig. 5a). Long-tailed miR-27a isomiRs that were absent when co-expressed with AGO2-F2L3 in WT cells became detectable in DIS3L2-KO cells (Fig. 5a). Deep sequencing analysis confirmed that the long miR-27a tails that accumulate upon DIS3L2 depletion were due to oligouridylation (Fig. 5b, Supplementary Fig. 5b). Together, these results strongly suggest that oligouridylated miR-27a was targeted by DIS3L2 for degradation. Long-tailed miR-27a isomiRs were absent in both WT and DIS3L2-KO cells when miR-27a was co-expressed with AGO2-WT (Fig. 5a, b, Supplementary Fig. 5b), confirming that an exposed 3′ end is required for a miRNA to be oligouridylated. The same experiments performed with miR-23a-3p and miR-134-5p generated similar results (Supplementary Fig. 5c, d), suggesting that DIS3L2 degrades oligouridylated mature miRNAs regardless of their sequence.

Given that DIS3L2 constantly removes oligouridylated miRNAs in cells, we revisited the characteristics of miRNA tailing enzymes without the interference of DIS3L2. To avoid having to knockout DIS3L2 in various TUT KO cell lines, we repressed DIS3L2 activity by co-expressing its catalytic-dead mutant

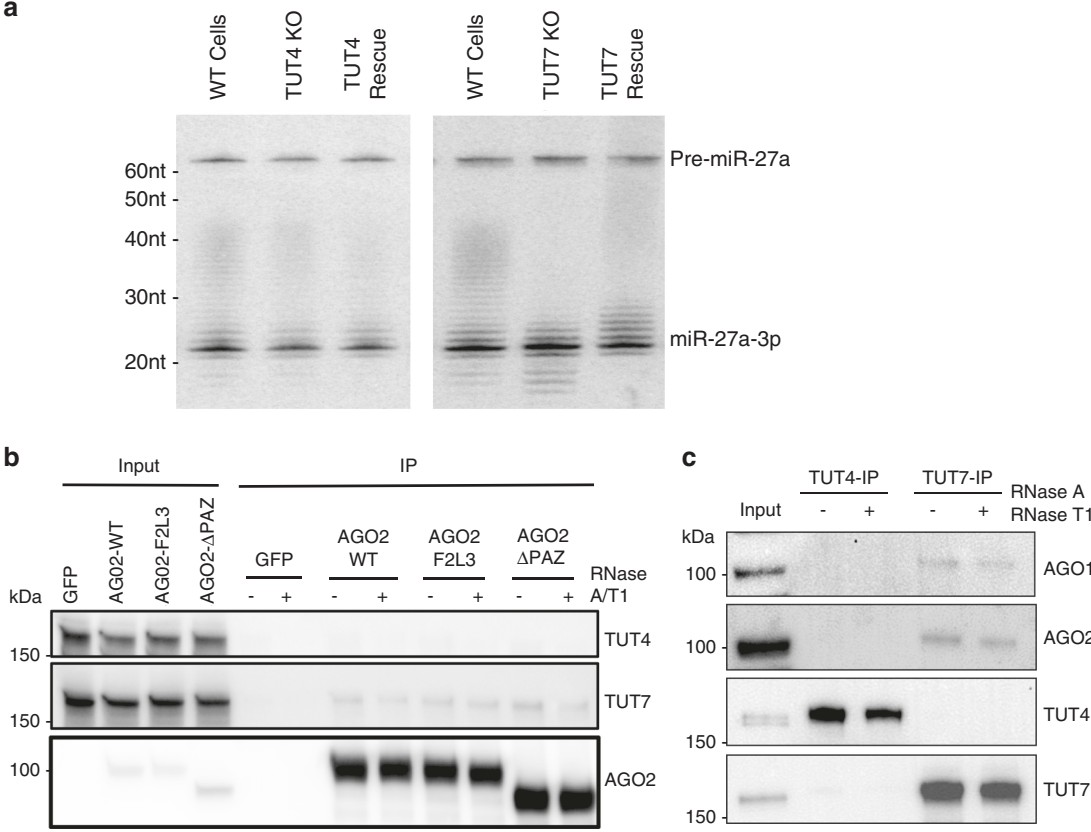

**Fig. 4 TUT7 associates with Argonaute proteins. a** Co-expression of pri-miR-27a and AGO2-ΔPAZ in wild-type, TUT4 KO and TUT7 KO cells with or without rescue. Detection of miR-27a-3p by Northern blot. **b** HEK293T cells were transfected with GFP, AGO2-WT, AGO2-F2L3, or AGO2-ΔPAZ. FLAG-tagged GFP or AGO2 was immunoprecipitated (IP) in the presence or absence of RNasesA/T1. Detection of TUT4, TUT7 and AGO2 by Western blot in input and FLAG-immunoprecipitate. **c** Endogenous TUT4 or TUT7 was immunoprecipitated by specific antibodies from HEK293T cells in the presence or absence of RNases. Detection of AGO1, AGO2, TUT4, and TUT7 by Western blot in input and immunoprecipitate. Source data are provided as a Source Data file.

(DIS3L2-CD-mut), which functions in a dominant-negative manner[39,40]. DIS3L2 inhibition resulted in the detection of long U-tails on miR-27a in TUT7 KO cells (Fig. 5c, d, Supplementary Fig. 5e), indicating that enzymes other than TUT7 contribute to oligouridylation of miRNAs. Further depletion of TUT4 (comparing TUT7 KO with TUT4/7 DKO) reduced the oligouridylation (Fig. 5c, d, Supplementary Fig. 5e), demonstrating the contribution of TUT4. The fact that oligouridylated miR-27a isomiRs were still detectable in TUT4/7 DKO cells suggests that additional enzymes can carry out oligouridylation (Fig. 5c), demonstrating extensive redundancy in the enzymes that tail AGO-bound miRNAs in cells. These conclusions were further supported by experiments performed with AGO2-F2L3 (Supplementary Fig. 5f) and miR-23a (Supplementary Fig. 5g).

To test whether DIS3L2 removes oligouridylated endogenous miRNAs, we transfected AGO2-WT, AGO2-F2L3, and AGO2-ΔPAZ in wild-type cells or the corresponding DIS3L2 KO cells, and then performed immunoprecipitation. We analyzed endogenous miRNAs in the pull-downs by deep sequencing. Consistent with the idea that the 3′ ends of miRNAs are protected by the AGO2-WT and become accessible when associated with AGO2 PAZ mutants, we observed accumulation of oligo-U-tails in the latter, but not in the former, upon DIS3L2 depletion (Fig. 5e, Supplementary Fig. 5h). Together, these results support a model in which AGO-bound miRNAs with an exposed 3′ end are oligouridylated and subsequently removed by DIS3L2.

**Redundant miRNA decay via tailing-independent 3′ trimming.** Abolishing oligouridylation by depleting TUT4 and TUT7 led to increased trimming (Fig. 3c, d, Supplementary Fig. 3d). Consistent with the idea that TUT7 is more robust in tailing mature miRNAs, knocking out TUT7 by itself was sufficient to trigger the accumulation of trimmed isomiRs (Fig. 4a, Supplementary Fig. 4c), and this was rescued by the expression of TUT7 (Fig. 3c, d, Supplementary Fig. 3c, d). These results suggest that tailing enzymes compete with trimming enzyme(s) in accessing the 3′ end of miRNAs. Depleting DIS3L2 or inhibiting DIS3L2 activity had no impact on the intensity of these trimmed isomiRs (Fig. 5a, Supplementary Fig. 5f, g). These results indicate that the trimming process is likely independent of DIS3L2, which is consistent with the observation that DIS3L2 prefers oligouridylated substrates[38,41]. Blocking oligouridylation had marginal effects on the level of miRNAs without 3′ modifications [see intensity of miRNA bands of canonical size (Figs. 3c and 4a, Supplementary Figs. 3d and 4b, c)], supporting a model in which 3′ oligouridylation and 3′ trimming function redundantly to degrade AGO-bound miRNAs.

Next, we deep sequenced endogenous miRNAs associated with AGO PAZ mutants in wild-type and TUT4/7 DKO cells. Although blocking TUT4 and TUT7 impaired oligouridylation, it had no detectable effect on the overall oligo-tailing (uridylation and adenylation) of endogenous miRNAs (Supplementary Fig. 5i), partially due to the fact that DIS3L2 constantly removes oligouridylated miRNAs. This also demonstrates the redundant contributions of other tailing enzymes. Nonetheless, knocking out

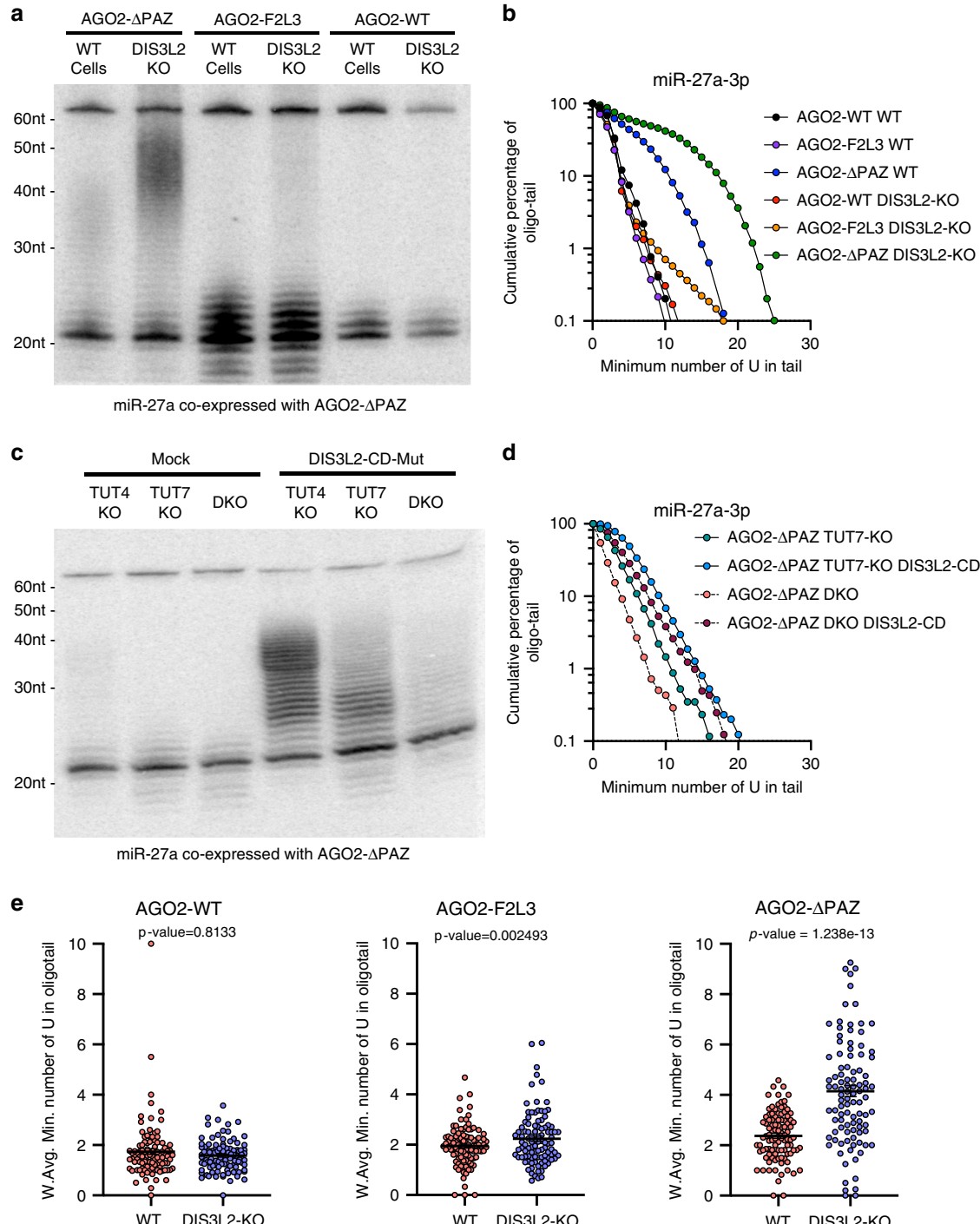

**Fig. 5 DIS3L2 degrades oligouridylated mature miRNAs. a** Co-expression of pri-miR-27a and FLAG-AGO2 constructs (WT, F2L3, and ΔPAZ) in HEK293T wild-type and DIS3L2 KO background. Detection of miR-27a-3p in input and FLAG-immunoprecipitate (IP) by Northern blot. **b** Composition of miR-27a-3p oligo-tail, number of U nucleotides in tail (x-axis) and cumulative percentage (y-axis), comparing AGO2 constructs (WT, F2L3, and ΔPAZ) in wild-type and DIS3L2 KO backgrounds. **c** Co-expression of pri-miR-27a, AGO2-ΔPAZ, and mock or DIS3L2 catalytic-dead mutant (DIS3L2-CD-mut) in TUT4 and TUT7 single and double KO cells. **d** Composition of miR-27a-3p oligo-tail, number of U nucleotides in tail (x-axis) and cumulative percentage (y-axis), comparing DIS3L2-CD-mut U-tail protection in TUT7 KO and TUT4/7 DKO cells. **e** Weighted average number of oligo-U-tail nucleotides added to endogenous miRNA loaded into AGO2 (WT, F2L3, and ΔPAZ) in wild-type and DIS3L2 KO backgrounds (mean ± SEM, N = 100 miRNAs). Wilcoxon's test for paired values (two-sided) were used to calculate the p values. Source data are provided as a Source Data file.

TUT4 and TUT7 led to a subtle increase in 3′ trimming (Supplementary Fig. 5j). Together, these results suggest that, besides the TUT-DIS3L2 machinery, a tailing-independent trimming process functions redundantly in degrading AGO-bound mature miRNAs from the 3′ end.

**TUT-DIS3L2 is implicated in but not required for TDMD.** To identify endogenous miRNAs targeted by the TUT-DIS3L2 machinery, we sequenced miRNAs in wild-type and DIS3L2 KO cells and compared the tail composition for each miRNA. Upon DIS3L2 depletion, the average numbers of U in tails

increased for miR-7-5p, miR-222-3p, and miR-769-5p, among others (Fig. 6a). Further analyses confirmed that only oligo-U-tails and, to a much lesser extent, oligo-A-tails accumulated (Fig. 6b, Supplementary Fig. 6a). Knocking out TUT4 and TUT7 had a marginal impact on the oligouridylation of miR-222-3p and miR-769-5p and no observable effect on miR-7-5p (Fig. 6b), suggesting that the majority of isomiRs oligouridylated by TUT4 and TUT7 are absent in wild-type cells, presumably removed by DIS3L2. These results were validated by Northern blot. Extensive tailed-isomiRs of endogenous miR-7-5p, miR-222-3p and miR-769-5p accumulated in DIS3L2 KO cells (Fig. 6c). As expected, knocking out TUT4 and TUT7 together with DIS3L2 KO abolished the extensive tailed-isomiRs (Fig. 6c), confirming that TUT4 and TUT7 are the main enzymes uridylating mature miRNAs. As a control, miR-21-5p did not gain U-tails upon DIS3L2 depletion (Fig. 6a, c). Together, these results support a model in which most endogenous miRNAs have their 3′ ends protected by the AGOs whereas a subset of miRNAs, including miR-7-5p, miR-222-3p, and miR-769-5p, have their 3′ ends exposed and are targeted by the TUT-DIS3L2 machinery.

It is possible that miR-7-5p, miR-222-3p, and miR-769-5p have their 3′ ends exposed because they are under active TDMD regulation in HEK293T cells. To test this, we sought to identify potential endogenous TDMD triggers. These triggers should be relatively abundant transcripts that contain target sites with the potential to extensively base-pair with the corresponding miRNA[42]. We used the calculated binding energy between miRNA and its predicted target site (seed-paired) as an indicator of the extent of base-pairing and measured the expression level of endogenous transcripts by RNA-seq. Analysis of miR-7-5p identified a known TDMD trigger—lncRNA CYRANO[17] (Supplementary Fig. 6b), validating this approach. While many potential TDMD triggers were identified for miR-222-3p and miR-769-5p, few were found for other highly expressed miRNAs such as miR-21-5p, miR-10a-5p, and miR-148a-3p (Supplementary Fig. 6b). This could explain why miR-7-5p, miR-222-3p, and miR-769-5p were targeted by the TUT-DIS3L2 machinery in HEK293T cells. Consistent with this idea, knocking down CYRANO using two independent siRNAs (Supplementary Fig. 6c) resulted in upregulation of miR-7 and reduced 3′ tailing and trimming, whereas overexpressing CYRANO, but not CYRANO with a mutated miR-7 binding site (CYRANO-mut) caused the opposite (Fig. 6d). Deep sequencing these samples confirmed that oligo-U tails but not oligo-C or oligo-G tails correlated with the TDMD effect (Supplementary Fig. 6d). Consistent with a previous study[17], oligo-A tailed forms of miR-7-5p also accumulated during TDMD (Fig. 6b, Supplementary Fig. 6d), suggesting that adenylation enzymes are also involved. Taken together, these results suggest that the TUT-DIS3L2 machinery is a part of the TDMD pathway.

Although miR-7-5p is targeted by the TUT-DIS3L2 machinery in HEK293T cells, miR-7-5p did not accumulate in cells depleted of TUT4/7 or DIS3L2 (Fig. 6c). Neither did miR-222-3p nor miR-769-5p (Fig. 6c). In fact, the level of miR-7-5p decreased in DIS3L2 KO cells, possibly due to an approximately threefold increase in CYRANO levels in DIS3L2 KO cells (Supplementary Fig. 6e) and/or an indirect effect of DIS3L2 depletion. Supporting this idea, levels of both U6 snRNA and Tyr-tRNA, were also reduced in the DIS3L2 KO cells (Supplementary Fig. 6f). Overexpressing CYRANO induced a similar degree of miR-7-5p decay in wild-type, TUT4/7 DKO and DIS3L2 KO cells (Fig. 6e). The reduction of canonical miR-7-5p was somewhat attenuated upon TUT4/7 or DIS3L2 depletion based on the Northern blot quantification (Supplementary Fig. 6g). Nonetheless, these results suggest that the TUT-DIS3L2 machinery is not essential for TDMD.

It is possible that exonucleases other than DIS3L2 function in parallel during TDMD by a 3′ trimming process that is independent of uridylation. To test this, we knocked down PARN, an A-tail-specific exonuclease, or EXOSC3, a key component of the cytoplasmic exosome complex, in both HEK293T cells and DIS3L2 KO cells > 90% by siRNAs (Supplementary Fig. 6h). Consistent with a previous study[17], knocking down PARN had marginal, if any, impact on the level of miR-7-5p. On the other hand, depletion of EXOSC3 by two independent siRNA sequences led to accumulation of trimmed miR-7-5p isoforms with or without DIS3L2 (Supplementary Fig. 6i). This effect is specific to miR-7-5p which is under active TDMD regulation (Supplementary Fig. 6i), suggesting that the exosome functions independently of the TUT-DIS3L2 machinery in TDMD. Quantification of Northern blot results of three biological replicates revealed a consistent but subtle increase of miR-7-5p relative to the miR-21-5p control upon exosome knocking down (Supplementary Fig. 6j), suggesting a redundant role of other exonuclease(s) besides DIS3L2 and exosome.

## Discussion

Although there is increasing evidence that 3′ non-templated tails are involved in miRNA turnover[7], the underlying mechanism remains elusive. Here, by creating AGO2 mutants carrying mutations at the PAZ domain, we studied how miRNA 3′ end modifications lead to decay. We provide evidence that AGO-bound mature miRNAs with an exposed 3′ end undergo oligo-tailing by TENTs including TUT4 and TUT7 and subsequent DIS3L2 degradation. Given that DIS3L2 prefers uridylated substrates[38,41], our data explain why miRNA uridylation is associated with instability whereas adenylation often stabilizes miRNAs[7]. Our results also suggest a tailing-independent 3′ trimming process which functions in parallel to degrade miRNAs, since blocking oligo-tailing or DIS3L2 function by itself does not abolish miRNA decay. Nonetheless, the finding of increased oligouridylated isomiRs in DIS3L2 knock-out cells reveals that a set of endogenous miRNAs are targeted by the TUT-DIS3L2 machinery. Together, our findings support a model in which the stability of miRNAs is controlled, at least in part, by the accessibility of their 3′ ends. The TUT-DIS3L2 pathway, together with other nucleases, efficiently removes AGO-bound miRNAs with an accessible 3′ end (Fig. 7). In plants, the redundancy between uridylation and trimming in miRNA decay, the AGO-TENT interaction, and the requirement of AGO for miRNA uridylation have been previously reported[43–46]. Our results both extend these observations and reveal that this is a conserved mechanism for controlling miRNA stability in both plants and animals. Our findings also lay the foundation for future studies to identify additional factors involved in miRNA decay.

Uridylation by TENTs in concert with DIS3L2 regulates the stability of a wide range of RNAs. TUT4 and TUT7 were initially identified as regulators of miRNA biogenesis at the precursor level[31–33]. Recently, TENTs-DIS3L2 has been identified as a cytoplasmic RNA surveillance pathway that degrades de-adenylated mRNAs[47], defective pre-miRNAs[48], mirtron-precursors[49,50], aberrant structured non-coding RNAs[51–53], unprocessed tRNAs[54], yeast Ago1-associated small RNAs[55], and rRNAs[56]. Together with previous reports that TUT4 and TUT7 uridylate miRNAs[34,35,57], our findings establish this machinery in miRNA decay as well. The majority of miRNAs have their 3′ ends protected by AGOs, leaving only a small set of miRNAs targeted by DIS3L2 in HEK293T cells. Therefore, the TUT-DIS3L2 machinery is unlikely to be a house-keeping pathway for miRNA turnover. Rather, it may function in situations that require rapid changes in miRNA abundance.

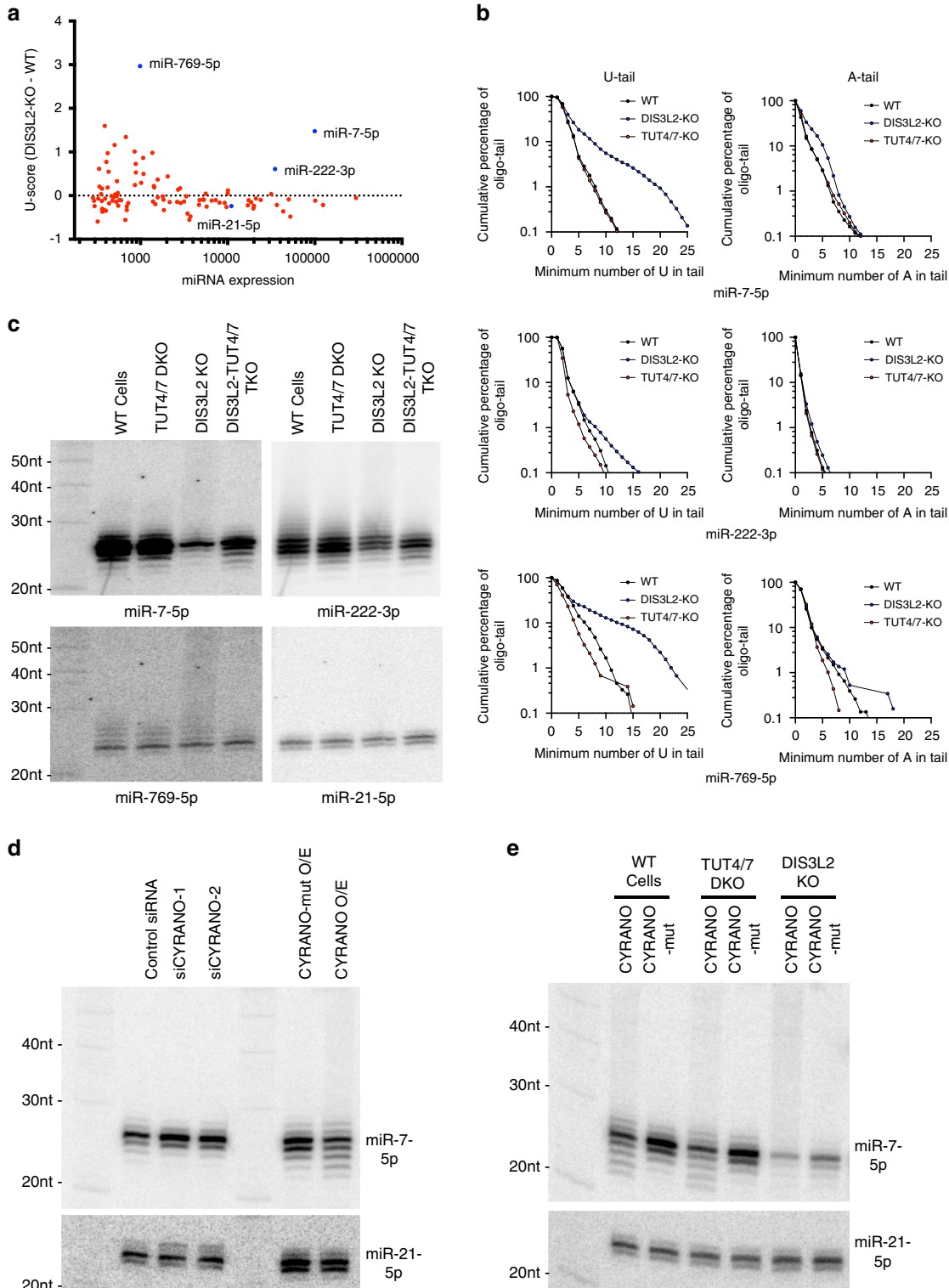

**Fig. 6 TUT-DIS3L2 is implicated in but not required for TDMD. a** Scatter plot of U-score (change in average number of oligo-U between DIS3L2 KO and wild-type cells, y-axis) and expression of endogenous miRNA (in counts per million, x-axis). Highlighted in blue, miR-7-5p, miR-222-3p, miR-769-5p, and miR-21-5p (control miRNA). **b** Comparing the composition of oligo-tail (number of U or A in the tail, x-axis) and cumulative percentage (y-axis) of miR-7-5p, miR-222-3p, and miR-769-5p, in wild-type, TUT4/7 DKO and DIS3L2 KO cells. **c** Detection of tailing of endogenous miR-7-5p, miR-222-3p, and miR-769-5p in wild-type, TUT4/7 DKO, DIS3L2 KO, and TUT4/7/DIS3L2 Triple KO (TKO) by Northern blot. miR-21-5p was also detected as a control. **d** Northern blot detecting endogenous miR-7-5p and miR-21-5p (control miRNA) in HEK293T cells upon knockdown or overexpression of CYRANO by siRNA (two independent sequences) or plasmids with miR-7 binding site (CYRANO) or mutated site (CYRANO-mut). **e** Northern blot detecting endogenous miR-7-5p and miR-21-5p (control miRNA) in wild-type, TUT4/7 DKO and DIS3L2 KO cells transfected with plasmids expressing CYRANO or CYRANO-mut. Source data are provided as a Source Data file.

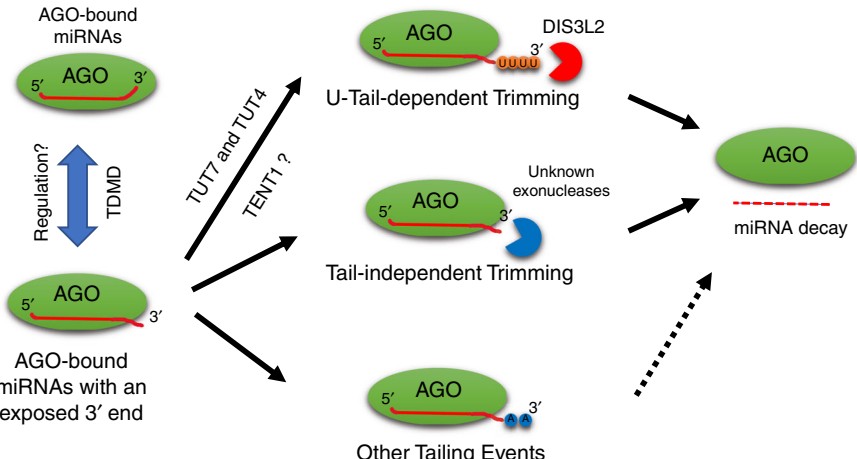

**Fig. 7 Proposed model of mature miRNA degradation via 3′ modifications.** The 3′ ends of miRNAs are protected by the AGO PAZ domain and become accessible upon specific regulations such as TDMD. Exposed miRNA 3′ ends are subjected to 3′ modifications, including (1) oligouridylation which leads to subsequent degradation by DIS3L2, (2) tail-independent trimming which results in decay, and (3) other tailing events such as adenylation which may or may not destabilize miRNAs.

Among the 11 TENTs identified in mammals, TENT1 (TUT1) in addition to TUT4 and TUT7 is implicated in uridylation of mature miRNAs[21,22]. The residual uridylation observed in cells depleted of TUT4 and TUT7 is likely due to TENT1 activity. In parallel, TENT2 (TUT2/GLD2/PAPD4), TENT4A, and TENT4B are involved in adenylation of miRNAs[21,22]. A recent report demonstrated that miRNAs with a 3′ A-tail are also degraded by DIS3L2[58]. These data indicate that miRNA 3′ modification is a complex and dynamic process and there are multiple mechanisms degrading miRNAs in parallel. This extensive redundancy at the level of tailing, trimming and miRNA decay highlights the importance of controlling miRNA stability via 3′ modifications while presenting a challenge in establishing the function of individual pathways. Future studies illustrating the specificity of various TENTs will aid our understanding of how these pathways are coordinated to regulate miRNA turnover.

Our findings implicate the TUT4/7-DIS3L2 machinery in TDMD, consistent with a previous report that DIS3L2 and TENT1 co-purify with TDMD targets[59]. However, knocking out DIS3L2 or TUT4 and TUT7 does not abolish CYRANO-induced TDMD of miR-7 in HEK293T cells, suggesting that there are redundant pathways, possibly a uridylation-independent 3′ trimming process mediated by exonucleases other than DIS3L2. Supporting this idea, inhibition of exosome function led to a subtle attenuation of TDMD efficiency and accumulation of trimmed forms of miR-7-5p isomiRs. It is possible that the exosome degrades trimmed isomiRs that disassociate from AGO. It is also possible that miRNAs are degraded by a means that is independent of 3′ modifications. Since TDMD of a given miRNA can be induced by multiple target sequences, it is intriguing to hypothesize that the underlying mechanism is dependent on the nature of its trigger.

Nearly all endogenous miRNAs are subjected to 3′ modifications[20], indicating that their 3′ ends are accessible at some point. However, only a small set of miRNAs are targeted by the TUT-DIS3L2 machinery. It is possible that a dislocated 3′ end, after being modified, falls back into the AGO PAZ domain, preventing further modifications. The TUT-DIS3L2 machinery is only implicated when the miRNA 3′ end is exposed for a prolonged amount of time. We have identified several AGO mutations in the PAZ domain in cancer patients which are correlated with increased levels of 3′ modifications. These mutations may contribute to cancer progression by altering the half-life of certain

miRNAs. Likewise, dysregulation of miRNA stability may contribute to phenotypes associated with TUT or DIS3L2 deficiency[60]. We demonstrated previously that 3′ mono-uridylation alters the way in which miRNAs interact with their targets[35]. Here, we find that 3′ oligouridylation leads to miRNA decay by DIS3L2. Future studies will address how various 3′ modifications and the resulting 3′ isomiRs coordinate in regulating miRNA function and stability.

## Methods
**Plasmids and siRNA.** For miRNA expression plasmids, the genomic sequence of miRNA and its flanking region (~250 bp on each side) was cloned into a CMV (Pol II) driven expression vector. The AGO2-F2L3, AGO2-D597A, and AGO2-ΔPAZ plasmids were generated by mutagenesis of pIRESneo-FLAG/HA AGO2 (Addgene, #10822)[61]. The coding sequence of DIS3L2 was polymerase chain reaction (PCR) amplified from a pool of HEK293T cDNAs and then cloned into pIRESneo-FLAG/ HA at NheI and EcoRI sites using In-fusion HD kit (Clontech). The DIS3L2-CD-mut (catalytic-dead mutant) plasmid was generated by point mutation of D391N on pIRESneo-FLAG/HA-DIS3L2 using Q5 Site-Directed Mutagenesis Kit (NEB). CYRANO and CYRANO-mut expression constructs were generated by cloning the synthesized dsDNA oligos (gblocks, IDT) into the psiCHECK2 vector between XhoI and SpeI sites using In-fusion HD kit (Clontech). siRNAs against CYRANO were referenced from Kim et al.[62] and synthesized by IDT. siRNAs against EXOSC3 were a gift from Dr. Sandra Wolin's Lab[63]. siRNAs against PARN were purchased from Dharmacon. Sequences of oligos and siRNA were listed in Supplementary Table 1.

**Cell culture and transfection.** HEK293T cells were purchased from ATCC (CRL-11268) and were maintained in high glucose DMEM (Thermo Fisher Scientific) supplemented with 10% FB Essence (VWR), 1× GlutaMAX™-I (Thermo Fisher Scientific), 1× MEM Non-Essential Amino Acids (Thermo Fisher Scientific), and 100 U/ml penicillin and 100 μg/ml streptomycin mixture (Thermo Fisher Scientific). For plasmid and siRNA transfection, Lipofectamine 3000 and Lipofectamine RNAiMAX were used, respectively, according to the manufacturer's protocols. TUT4 KO, TUT7 KO, and DIS3L2 KO cell lines were established by transfecting LentiCRISPR V2-sgRNA with puromycin resistant marker into HEK293T cells. DIS3L2-TUT4/7 TKO cell line was established by transfecting DIS3L2KO cells with TUT4 and TUT7 LentiCRISPRV2-sgRNAs with hygromycin and zeocin resistant marker, respectively. Single colonies were picked and screened after drug selection.

**Western blot.** Total proteins were extracted by lysing the cells in the modRIPA buffer (10 mM Tris-Cl pH 7.0, 150 mM NaCl, 1 mM EDTA, 1% Triton X-100, and 0.1% sodium dodecyl sulfate (SDS)) supplemented with protease inhibitor cocktail (Roche). Proteins were resolved on 4–15% Mini-PROTEAN® TGX stain-free™ protein gels (Bio-Rad) and transferred onto a PVDF membrane using Trans-Blot Turbo Transfer System (Bio-Rad) according to the manufacturer's instructions. The primary antibodies used in this study were anti-DIS3L2 (1:1000, Sigma #HPA035797), anti-ZCCHC11 (1:500, Proteintech, #18980-1-AP), anti-ZCCHC6 (1:2000, Proteintech, #25196-1-AP), anti-Tubulin (1:3000, Sigma, #T9026),

anti-AGO1 (1:500, Wako, #015-22411), and anti-AGO2 (1:500, Wako, #015-22031). The signals were developed with SuperSignal West Pico Chemiluminescent Substrate (Pierce) for strong signals or Immobilon Western Chemiluminescent HRP Substrate (Millipore) for weak signals and imaged by the Chemidoc Touch Imaging System (Bio-Rad).

**Northern Blot**. Total RNA was extracted by using TRIzol reagent (Ambion). Twenty microgram total RNA and a $^{32}$P-labeled Decade marker (Ambion) were loaded into 20% (w/v) acrylamide/8 M urea gels. After gel electrophoresis, RNAs were transferred onto Hybond-N membranes (Amersham Pharmacia Biotech) using a semidry transfer apparatus, followed by either UV cross-linking using 1500J for detecting over-expressed miRNAs or EDC (1-ethyl-3-(3-dimethylami-nopropyl) carbodiimide)-mediated chemical cross-linking (Sigma) at 60 °C for 1 h for detecting endogenous miRNAs. $^{32}$P-labeled probes were hybridized with membrane overnight at 37 °C after pre-hybridized with PerfectHyb™ Plus Hybridization Buffer (Sigma) at 37 °C for 10 min. After washing with 2× SSC plus 0.1% SDS buffer for 3 × 15 min at 37 °C, the membrane was exposed to an Imaging Screen-K (Bio-Rad) overnight. Images were then analyzed by Typhoon Trio Imaging System (GE Healthcare). Northern blot results were quantified by Quantity One (Bio-Rad). The sequences of probes used in this study were listed in Supplementary Table 1.

**RNA immunoprecipitation**. For isolating FLAG-tagged AGO2-bound miRNAs, one 10-cm dish of HEK293T cells transfected with AGO2-WT, AGO2-F2L3, or AGO2-ΔPAZ, with or without miRNA expression vector, were lysed in 1 ml modRIPA buffer (10 mM Tris-Cl pH 7.0, 150 mM NaCl, 1 mM EDTA, 1% Triton X-100, and 0.1% SDS) supplemented with proteinase inhibitor cocktail (Roche). Cell lysates were incubated with 50 μl Anti-FLAG M2 Magnetic Beads (Sigma) at 4 °C overnight with rotation. For immunoprecipitation of endogenous AGO2-bound RNAs, HEK293T, DIS3L2 KO, and TUT4/7 DKO cells were collected and lysed with NP-40 buffer (50 mM HEPES-KOH [pH 7.4], 150 mM KCl, 2 mM EDTA, 0.5 mM DTT, 0.5% NP-40, complete EDTA-free protease inhibitor cocktail), 1 mM NaF, 5% [v/v] glycerol). The lysates were incubated with anti-AGO2 monoclonal antibody (Wako, #015-22031) conjugated to protein G magnetic beads (Bio-Rad, #1614023) for 1 h at RT with rotation. After five washes with BC150 buffer (20 mM Tris-HCl (pH 8.0), 150 mM KCl, 0.2 mM EDTA, 10% glycerol) at room temperature, the beads were lysed in 1 ml Trizol (Life Technologies) for RNA extraction.

**Protein co-immunoprecipitation (co-IP)**. For FLAG co-IP, FLAG-tagged AGO2-WT, AGO2-F2L3, or AGO2-ΔPAZ were transfected into HEK293T cells. Forty-eight hour after transfection, cells were lysed in modRIPA buffer. Cell lysates were cleared by centrifugation at 20,000g for 15 min at 4 °C. Totally, 5% supernatants were saved as input and the remaining was incubated with pre-washed Anti-FLAG M2 magnetic beads (Sigma #M8823) at 4 °C. The beads were washed with TBS buffer (50 mM Tris-Cl, PH 7.4, and 150 mM NaCl) for five times and then divided into two tubes. One tube was added with protein loading buffer (RNase−) and the other tube was treated with 1 μl RNase A (Thermo Scientific #EN0531) and 1 μl RNase T1 (Thermo Scientific #EN0542) for 20 min at room temperature (RNase+). The IP samples were loaded to the protein gel as described above and detected with anti-ZCCHC11 (1:500, Proteintech #18980-1-AP), anti-ZCCHC6 (1:2000, Proteintech #25196-1-AP), and anti-AGO2 (1:500, Wako, #015-22031). For the endogenous protein interaction, HEK293T cells were collected and co-IP was performed using either anti-ZCCHC11 or anti-ZCCHC6 to pull down endogenous TUT4 or TUT7, treated with or without RNase A/T1, and then Western blot was done to detect AGO1, AGO2, TUT4, and TUT7.

**Real-time PCR**. Five microgram total RNA was used for reverse transcription by SuperScript IV reverse transcriptase (Thermo Fisher Scientific) and random hexamers according to the manufacturer's instructions. 4 μl diluted cDNA (1:8) was used for real-time qPCR using iQ SYBR green supermix (Bio-Rad) in 10 μl reaction. Data were collected in CFX384 Touch Real-time PCR detection system (Bio-Rad). PCR primers used in this study were listed in Supplementary Table S1.

**Small RNA sequencing**. Small RNA libraries were constructed by NEBNext® small RNA library preparation kit (NEB, E7330) according to the manufacturer's protocol with minor modifications. In particular, 1 μg total RNA or 300 ng of AGO2-IP RNA was ligated to the 3′ adapter at a lower temperature (16 °C), with a higher PEG concentration (20%) and for a longer time (18 h). The cDNAs were then PCR amplified for 12–15 cycles, and the amplified library was purified by running on a 6% (w/v) native acrylamide gel with a 20-bp ladder. The 140–160 bp fraction of the library was excised from the gel, and then purified by ethanol precipitation. The small RNA library quality was assessed on the Agilent 2100 Bioanalyzer (Agilent), and the quantity was determined by Qubit dsDNA HS Assay (Life Technologies). Each small RNA library was sequenced on an Illumina MiSeq platform (Illumina) with MiSeq® Reagent Kit v3-150 cycle (Illumina).

**Analysis of small RNA sequencing data and tail composition**. The small RNA sequencing data were analyzed using an in-house pipeline. Briefly, adapters were removed, reads were mapped using Bowtie and visualized using IGV. More detailed study of the isomiR profile was done using QuagmiR[64]. This software uses a unique algorithm to pull specific reads and aligns them against a consensus sequence in the middle of a miRNA, allowing mismatches on the ends to capture 3′ isomiRs. The reports included tabulated analysis of miRNA expression, length, number of nucleotides trimmed and tail composition at individual read level. Customized R scripts were used to calculate percentages of canonical miRNA (defined as the most abundant templated read) and 3′ isomiRs, as well as percentages of tailing and trimming. Long tail composition was calculated by counting the number of non-templated nucleotides present in the tail of each isomiR read. Reads with equal number of non-templated nucleotides in the tail were added together and cumulative distribution was calculated for all the oligo-tailed isomiRs going from ones with longer to shorter tails. Script of the R code used to generate this tail composition analysis is available at GitHub.

**Analysis of isomiR profiles on AGO1 and AGO2 from TCGA**. Tumoral samples from TCGA bearing genomic mutations in either AGO1 or AGO2 leading to missense and synonymous amino acid changes were identified from Genomic Data Commons Data Portal (accessed during May 2019). GDC uses combined reports from several variant callers (mutect2, varscan, muse and somaticsniper). Selected Case ID were: P295L TCGA-53-A4EZ, R315M TCGA-HU-A4G8; and E299K TCGA-Z6-A8JE (AGO2), F310L TCGA-94-7033 (AGO1). The analysis of selected patient samples was also performed using QuagmiR[64], with a previous conversion of the bam files to fastq files by Picard Sam-to-Fastq, using Amazon cloud instances through the Seven Bridges Genomics implementation of the NCI Cancer Genomics Cloud. Script of the R code used to analyze the impact of AGO mutations is available at GitHub. Mutations were plotted into the PDB structures of AGO1 and AGO2 using pymol.

**Bioinformatic prediction of TDMD triggers**. Prediction of target RNAs with extensive 3′ pairing with miRNAs that could induce the dislocation of the 3′ end of the miRNAs from the PAZ domain was obtained by identifying RNAs with a 7mer seed from the list of human mRNA 3′UTRs (TargetScan7.2) and lncRNA (LNCipedia). Next, for each miRNA-RNA pair their hybridization minimum free energy (MFE) was calculated using RNAduplex from the ViennaRNA Package 2.0[65]. MFE for each miRNA-RNA hybrid was plotted against the abundance of the target RNA in HEK293 cells[35]. Script of the R code used to predict TDMD triggers is available at GitHub.

**Statistics and reproducibility**. Wilcoxon's test (two-sided) for paired values were used to calculate the $p$ values for the same group of miRNAs between two conditions or treatments. Otherwise, unpaired Student's $t$ test (two-sided) was used for comparing two groups. Error bars displayed on graphs represent the mean ± SEM of at least three independent experiments. All Northern blot and Western blot results showed here are representative results from at least three independent experiments.

**Reporting summary**. Further information on research design is available in the Nature Research Reporting Summary linked to this article.

## Data availability

The data that support this study are available from the corresponding author upon reasonable request. Sequence data that support the findings of this study have been deposited in GEO with the accession codes GSE139567 and GSE121327. The source data underlying Figs. 1a, 1c, 3a, 2a–d, f, 3b–d, 4a–c, 5a–e, 6a–e, and Supplementary Figs. 1d, 2a, d, f, 3a–d, 4a–c, 5a–j, 6a–j are provided as a Source Data file, which is also available at Mendeley (https://doi.org/10.17632/s5hss3jw6k.2).

## Code availability

Scripts of the R code used in this study are available at GitHub. Other code is available from the corresponding author upon reasonable request.

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

## Acknowledgements

We thank Dr. Sandra L. Wolin for critical reading of the manuscript and helpful discussions. This work has been funded by the intramural research program of the National Cancer Institute, National Institutes of Health (ZIA BC 011566). The Seven Bridges Cancer Genomics Cloud has been funded in whole or in part with Federal funds from the National Cancer Institute, National Institutes of Health, Contract no.

HHSN261201400008C and ID/IQ Agreement no. 17 × 146 under Contract no. HHSN261201500003I.

## Author contributions

A.Y., T.J.S., X.B.-D.R., and S.G. designed the research. A.Y. and T.J.S. performed the most experiments with helps from C.L., P.V., and L.D.; X.B.-D.R. did all the bioinformatic analyses; A.Y., T.J.S., X.B.-D.R., and S.G. analyzed the data. A.Y., T.J.S., X.B.-D.R., and S.G. wrote the paper.

## Competing interests

The authors declare no competing interests.
