## [Peer Review File · Nature Communications]

Reviewers' comments:

Reviewer #1 (Remarks to the Author):

This research studied miRNA 3' uridylation in human cell lines. The authors showed that mutations disrupting miRNA 3' binding of AGO2 causes miRNA 3' uridylation, which triggers miRNA degradation by DIS3L2. miRNA 3' uridylation is catalyzed by TUT4 and TUT7. Moreover, the authors also observed the presence of 3' trimming process that acts redundantly with uridylation to removing miRNAs with an exposed 3' end. The authors further showed that a subset of endogenous miRNAs in human HEK293T cells are targets of TUTs and DIS3L2. These results extend the observations in plants, and thus suggest that uridylation and trimming, which depends on AGO, are conserved mechanisms controlling miRNAs stability in both plants and animals. In general the manuscript are well written and experiments are well designed.

Two minor suggestions are:

In plants and animals, extensive base pairing release small RNA 3' end from the PAZ domain and causes their uridylation. It will be interesting to examine or discuss if endogenous miRNAs, which have potential extensive base-pairing targets in animals, are modified under wild-type cells (normal AGO)

The redundancy between uridylation and trimming, the AGO-TUT interactions (Current Biology 22:689-94, Current Biology, 22:695-700, PNAS 111:6365-6370), and the requirement of AGO on uridylation (The Plant Cell 25:2417-2428 ; PNAS 111:6365-6370) have been observed in plants. These should be properly discussed.

Reviewer #2 (Remarks to the Author):

Review of the manuscript: AGO-bound mature miRNAs with an exposed 3' end are oligouridylated by TUTs and subsequently degraded by DIS3L2

by Yang et al.

The authors present a study demonstrating the involvement of the TUT-DIS3L2 pathway in the degradation of miRNAs bound to AGO2 in human cells. By combination of protein immunoprecipitations and RNA analyses by northern blot and sequencing, this work shows that when miRNA 3' termini are not efficiently protected by AGO2, they become exposed for 3' terminal modification by TUT4 and TUT7 terminal uridylyltransferases. The oligo(U) tails are subsequently trimmed by DIS3L2. This its self is not a novel finding, however this work further indicates that TUT7 directly interacts with AGO2 and promotes formation of longer tailing. The authors have in addition identified a subset of miRNAs with oligo(A) tails

which do not appear to be formed and targeted by TUT-DIS3L2. In addition, AGO2 mutants in 3' miRNA protection accumulate truncated forms of miRNAs that are not primarily trimmed by DIS3L2. I find this work interesting, technically solid, however to my opinion it lacks enough novelty and contains loose ends. The following are the major concerns.

It is not shown, that the truncated miRNAs identified with AGO2-F2L3 and AGO2-deltaPAZ are consequence of TDMD. The TUT-DIS3L2 has been already several years ago demonstrated to play a role in the TDMD. This work shows the presence of DIS3L2 independent tails (oligoA) and trimming. The authors should show which TENT is responsible for oligo(A) tailing. They should also test whether the cytoplasmic exosome (the likely suspect) targets oligoadenylated miRNAs and/or the non-tailed miRNAs. These three major findings would in my opinion bring the novelty of the work worth considering publishable in NSMB.

I have several further comments:

1. On page 5, first paragraph they state: "Depp sequencing analysis confirmed that those isomiRs were mature miR-27a with variable lengths" I suggest to rather call them aberrant miRNAs as it is not clear whether they had been properly processed at first place. The same page, second paragraph sentence: Because the former but not the later is retained in mature RISC - there should be reference to the Supplementary Fig. 1c, not 1d.
2. This work is largely based on bioinformatics analyses of sequencing outputs of miRNA cDNA libraries and identification of the tailing status. This is not a trivial task. The methods section needs much more extensive and detailed description of the used algorithms and scripts for a reader to be able to judge whether the analysis was done correctly.
3. Figure 4B lack a negative control - for instance FLAG alone or FLAG-tagged GFP IP to demonstrate that TUT7 signal is specific in AGO2 IPs. Why is AGO2 not detected in the input samples?
4. The DIS3L2 KO cells display lower levels of some endogenous coprecipitated miRNAs, which is contra intuitive (e.g. Figure 6c, 6e). What is the interpretation of this result?

Reviewer #3 (Remarks to the Author):

The authors demonstrated a TUT-DIS3L2 machinery that mediated miRNA turnover, in which AGO-bound mature miRNAs with an exposed 3' end undergo oligo-tailing by TENTs including TUT4 and TUT7 and subsequent DIS3L2 degradation, through a series of overexpression and knock out experiments. This part of conclusion was well proven and also of important significance. The manuscript is well written, data are clearly presented and the methodology is in general sufficiently described.

1. However, if the TUT-DIS3L2 machinery participate in endogenous miRNA turnover, and its physiological significance, was still elusive.
2. Is it possible to study the TUT-DIS3L2 machinery in the AGO mutant patients representing in Fig2? It will be helpful for addressing its physiological effects.
3. In FigS3d, it seems that more trimming was occurred in AGO F2L3 mutant comparing to AGO Δ PAZ mutant. What is the possible mechanism?
4. What is the situation about the endogenous miR-27a in TUT DKO cells, not with the overexpression of pri-miR-27a?
5. In Fig4b, why AGO2 was absent in input?
6. FigS5f including the Mock control should be put in the main figures instead of Fig5c.
7. The endogenous control was absent for northern blot in Fig6c.
8. Nearly all endogenous miRNAs are subjected to 3' modifications, why miR-7-5p, miR-222-3p and miR-769-5p was specifically targeted by TUT4/7?
9. If the TUT-DIS3L2 machinery is not essential for TDMD. What is the physiological effects of TUT-DIS3L2 mediated miRNA turnover?

Reviewers' comments:

Reviewer #1:

This research studied miRNA 3' uridylation in human cell lines. The authors showed that mutations disrupting miRNA 3' binding of AGO2 causes miRNA 3' uridylation, which triggers miRNA degradation by DIS3L2. miRNA 3' uridylation is catalyzed by TUT4 and TUT7. Moreover, the authors also observed the presence of 3' trimming process that acts redundantly with uridylation to removing miRNAs with an exposed 3' end. The authors further showed that a subset of endogenous miRNAs in human HEK293T cells are targets of TUTs and DIS3L2. These results extend the observations in plants, and thus suggest that uridylation and trimming, which depends on AGO, are conserved mechanisms controlling miRNAs stability in both plants and animals. In general the manuscript are well written and experiments are well designed.

Thanks for the positive comments!

Two minor suggestions are:

In plants and animals, extensive base pairing release small RNA 3' end from the PAZ domain and causes their uridylation. It will be interesting to examine or discuss if endogenous miRNAs, which have potential extensive base-pairing targets in animals, are modified under wild-type cells (normal AGO)

This is a great suggestion! To identify these extensive basepairing targets expressed in wild-type HEK293T cells, we used the calculated binding energy between miRNA and its predicted target site (seed-paired) as an indicator of the extent of base-pairing and measured the expression level of endogenous transcripts by RNA-seq. Analysis of miR-7-5p identified the lncRNA CYRANO, which contains many extensively basepaired target sites of miR-7-5p. This validated our approach. While many extensive basepairing targets were identified for miR-222-3p and miR-769-5p, few were found for other highly expressed miRNAs such as miR-21-5p, miR-10a-5p and miR-148a-3p (Supplementary Fig 6b). This could explain why miR-7-5p, miR-222-3p and miR-769-5p had their 3' ends exposed when associated with the endogenous (normal) AGO2 and therefore were targeted by the TUT-DIS3L2 machinery in HEK293T cells.

The redundancy between uridylation and trimming, the AGO-TUT interactions (Current Biology 22:689-94, Current Biology, 22:695-700, PNAS 111:6365-6370), and the requirement of AGO on uridylation (The Plant Cell 25:2417-2428 ; PNAS 111:6365-6370) have been observed in plants. These should be properly discussed.

We apologize for these omissions. These papers were cited and these findings in plants were discussed in the revised manuscript.

Reviewer #2:

The authors present a study demonstrating the involvement of the TUT-DIS3L2 pathway in the degradation of miRNAs bound to AGO2 in human cells. By combination of protein immunoprecipitations and RNA analyses by northern blot and sequencing, this work shows that when miRNA 3' termini are not efficiently protected by AGO2, they become exposed for 3' terminal modification by TUT4 and TUT7 terminal uridylyltransferases. The oligo(U) tails are subsequently trimmed by DIS3L2. This itself is not a novel finding, however this work further indicates that TUT7 directly interacts with AGO2 and promotes formation of longer tailing. The authors have in addition identified a subset of miRNAs with oligo(A) tails which do not appear to be formed and targeted by TUT-DIS3L2. In addition, AGO2 mutants in 3' miRNA protection accumulate truncated forms of miRNAs that are not primarily trimmed by DIS3L2. I find this work interesting, technically solid, however to my opinion it lacks enough novelty and contains loose ends. The following are the major concerns.

It is not shown, that the truncated miRNAs identified with AGO2-F2L3 and AGO2-deltaPAZ are consequence of TDMD.

This is a great point. We agree that the trimming process might be different between miRNAs associated with AGO2-PAZ-mutants and miRNAs during TDMD because a target transcript presents in the latter but not in the former. We will mind this difference when searching trimmers involved in TDMD. Nonetheless, it is possible that mechanisms besides TDMD could trigger the exposure of miRNA 3' end, for example post-translational modifications of AGO2 or mutations at AGO2 PAZ domain. Our findings are relevant in these scenarios.

The TUT-DIS3L2 has been already several years ago demonstrated to play a role in the TDMD. This work shows the presence of DIS3L2 independent tails (oligoA) and trimming. The authors should show which TENT is responsible for oligo(A) tailing.

This is a great suggestion. Using CRISPR, we have knocked out TENT2 (GLD2), the known TENT adenylating mature miRNAs. The TDMD-induced oligo-A tail of miR-7 was not abolished in TENT2 KO cells. In fact, the TDMD-induced oligo-tail of miR-7 remains detectable by northern blot and NGS when TUT4, TUT7 and TENT2 were all depleted (triple KOs). It became clear that there are additional tailing enzymes involved. We therefore decided to not include this part of result in this manuscript which focuses on uridylation. We are actively pursuing this direction and hope to present a comprehensive study of miRNA adenylation in near future.

They should also test whether the cytoplasmic exosome (the likely suspect) targets oligoadenylated miRNAs and/or the non-tailed miRNAs. These three major findings would in my opinion bring the novelty of the work worth considering publishable in NSMB.

This is another great suggestion. Using siRNAs, we knocked down PARN, an A-tail-specific exonuclease, or EXOSC3, a key component of the cytoplasmic exosome complex, >90% in both HEK293T cells and DIS3L2 KO cells. We then measured the level of the TDMD-targeted miR-7 to assess the role of these two nucleases in TDMD (Supplementary Fig. 6i). Knocking down PARN had marginal impact, indicating that PARN is unlikely to be the nuclease targeting oligo-A tailed miRNAs during TDMD. On the other hand, depletion of EXOSC3 by two independent siRNA sequences led to a specific accumulation of trimmed miR-7-5p isoforms (Supplementary Fig. 6i). It is possible that exosome facilitates the removal of trimmed miRNAs during TDMD. Inhibiting exosome in DIS3L2 KO cells attenuated but not abolished the decay of miR-7, suggesting a redundant role of other exonucleases besides DIS3L2 and exosome. We have included in the revised manuscript this part of results, which provided additional insights into the tailing-independent trimming process of AGO-associated miRNAs.

I have several further comments:

1. On page 5, first paragraph they state: "Depp sequencing analysis confirmed that those isomiRs were mature miR-27a with variable lengths" I suggest to rather call them aberrant miRNAs as it is not clear whether they had been properly processed at first place. The same page, second paragraph sentence: Because the former but not the later is retained in mature RISC - there should be reference to the Supplementary Fig. 1c, not 1d.

Thanks for pointing this out. We have revised the manuscript accordingly.

2. This work is largely based on bioinformatics analyses of sequencing outputs of miRNA cDNA libraries and identification of the tailing status. This is not a trivial task. The methods section needs much more extensive and detailed description of the used algorithms and scripts for a reader to be able to judge whether the analysis was done correctly.

We apologize for not stating this clearly in the previous submission. We have created a webpage in GitHub where scripts of the R code used, and detailed description of analysis, are available. (<https://github.com/Gu-Lab-RBL-NCI/oligo-tail-miRNA>). Link to this webpage is included in the method section of the revised manuscript.

3. Figure 4B lack a negative control - for instance FLAG alone or FLAG-tagged GFP IP to demonstrate that TUT7 signal is specific in AGO2 IPs. Why is AGO2 not detected in the input samples?

Thanks for pointing this out. We have repeated the AGO2-TUT4/7 co-IP experiment with the suggested negative control (FLAG-GFP). As shown in the new Figure 4b, no TUT7 was detected in the FLAG-GFP pulldown, indicating that AGO2-TUT7 interaction is specific. Of note, the signal from endogenous AGO2 was too weak compared to that of overexpressed FLAG-AGO2, therefore was undetectable in this WB experiment.

Only 1% of the input was loaded in the previous WB. The AGO2 in input was missing due to insufficient exposure. We have increased the input to 5% and adjusted the exposure time to make sure that AGO2 signal was visible in the new Figure 4b. A longer exposure of the previous figure 4b is attached below. The first three lanes are input; the six lanes on the right are pulldowns.

4. The *DIS3L2* KO cells display lower levels of some endogenous coprecipitated miRNAs, which is contra intuitive (e.g. Figure 6c, 6e). What is the interpretation of this result?

This is a great point and a keen observation. We were also puzzled by this result. Given that *DIS3L2* was shown to regulate the metabolism of a wide range of RNAs, the reduced level of endogenous miRNAs is possibly due to an indirect effect of *DIS3L2* depletion. Supporting this idea, the levels of both U6 snRNA and Tyr-tRNA were reduced in the *DIS3L2* KO cells despite the fact that the same amount of RNA was loaded at each lane (Supplementary Fig. 6f). This is a representative result from three biological replicates. We have discussed this accordingly in the revised manuscript.

Reviewer #3:

The authors demonstrated a TUT-DIS3L2 machinery that mediated miRNA turnover, in which AGO-bound mature miRNAs with an exposed 3' end undergo oligo-tailing by TENTs including TUT4 and TUT7 and subsequent DIS3L2 degradation, through a series of overexpression and knock out experiments. This part of conclusion was well proven and also of important significance. The manuscript is well written, data are clearly presented and the methodology is in general sufficiently described.

Thanks for the positive comments!

1. However, if the TUT-DIS3L2 machinery participate in endogenous miRNA turnover, and its physiological significance, was still elusive.

The extensive redundancy at the level of tailing, trimming and miRNA decay makes it challenging to establish the explicit role of the TUT-DIS3L2 machinery in control miRNA turnover. Nonetheless, we provided compelling genetic and biochemical evidence that TUT-DIS3L2 degrades AGO-associated miRNAs with accessible 3' ends. While the physiological significance remains unclear, further investigation, for example the study of PAZ mutations identified in cancer patients, will provide additional insights. We have discussed this accordingly in the revised manuscript.

2. Is it possible to study the TUT-DIS3L2 machinery in the AGO mutant patients representing in Fig2? It will be helpful for addressing its physiological effects.

This is a great suggestion. We hypothesize that these mutations may contribute to cancer progression by altering the half-life of certain miRNAs: increasing stability of oncomiRs and/or promoting decay of tumor suppressor miRNAs. We are actively examining these possibilities and hope to report the result in a future study.

3. In FigS3d, it seems that more trimming was occurred in AGO F2L3 mutant comparing to AGO Δ PAZ mutant. What is the possible mechanism?

This is a great point and a keen observation. As tailing of miRNAs was less robust with AGO2-F2L3 than with AGO2- Δ PAZ, more trimming occurred in AGO2-F2L3-associated miRNAs aligns with the idea that tailing enzymes compete with trimming enzyme(s) in accessing the 3' end of miRNAs. It is possible that tailing enzymes (TUT4/7) are more sensitive to spatial constraints. Removal the PAZ domain to completely expose the miRNA 3' end is therefore more beneficial to the tailing process than to the trimming process.

4. What is the situation about the endogenous miR-27a in TUT DKO cells, not with the overexpression of pri-miR-27a?

miR-27a is barely expressed in HEK293T cells and the corresponding TUT4/7 DKO cells, making it extremely difficult to examine the status of its 3' end. Nonetheless, conclusions draw from the study of overexpressed miR-27a were confirmed with endogenous miRNAs. Northern Blots and NGS analyses presented in Figure 2, Figure 6 and the corresponding supplementary figures are experiments performed without overexpression of any pri-miRNAs.

5. In Fig4b, why AGO2 was absent in input?

Thanks for pointing this out. Reviewer2 had a similar concern. We have repeated this experiment with an additional negative control. Please see our response to reviewer2 for detail.

To your question, only 1% of the input was loaded in the previous WB. The AGO2 was absent in input due to insufficient exposure. A longer exposure of the previous figure 4b is attached below. The first three lanes are input; the six lanes on the right are pulldowns.

6. FigS5f including the Mock control should be put in the main figures instead of Fig5c.

Thanks for pointing this out. We have included the mock control in Fig5c as suggested.

7. The endogenous control was absent for northern blot in Fig6c.

Thanks for pointing this out. Since all miRNAs in Fig6c are endogenous miRNAs probed from the same blot, we reasoned that they could serve as controls to each other. Given that TUT and DIS3L2 were shown to regulate the metabolism of a wide range of RNAs, including Pol III transcripts (U6 snRNA for example), mRNAs, rRNAs and tRNAs, it is difficult to find a loading control for this particular northern blot. In fact, although same amount of RNA was loaded at each lane, the levels of both U6 snRNA and Tyr-tRNA were reduced in the DIS3L2 KO cells (Supplementary Fig. 6f). This is a representative result from three biological replicates.

8. Nearly all endogenous miRNAs are subjected to 3' modifications, why miR-7-5p, miR-222-3p and miR-769-5p was specifically targeted by TUT4/7?

This is a great point. Reivewer1 had a similar question. It is possible that miR-7-5p, miR-222-3p and miR-769-5p have their 3' ends exposed and therefore targeted by the TUT-DIS3L2 machinery because they are under active TDMD regulation in HEK293T cells. To test this, we sought to identify potential endogenous TDMD triggers. These triggers should be relatively abundant transcripts that contain target sites with the potential to extensively base-pair with the corresponding miRNA. We used the calculated binding energy between miRNA and its predicted target site (seed-paired) as an indicator of the extent of base-pairing and measured the expression level of endogenous transcripts by RNA-seq. Analysis of miR-7-5p identified a known TDMD trigger - lncRNA CYRANO (Supplementary Fig 6b), validating this approach. While many potential TDMD triggers were identified for miR-222-3p and miR-769-5p, few were found for other highly expressed miRNAs such as miR-21-5p, miR-10a-5p and miR-148a-3p (Supplementary Fig 6b). This could explain why miR-7-5p, miR-222-3p and miR-769-5p were targeted by the TUT-DIS3L2 machinery in HEK293T cells.

9. If the TUT-DIS3L2 machinery is not essential for TDMD. What is the physiological effects of TUT-DIS3L2 mediated miRNA turnover?

Our results indicate that once the 3' end of AGO-associated miRNA is exposed, oligo-uridylation by TUT4/7 and the subsequent removal by DIS3L2 are the predominant process. This suggests that other miRNA decay processes in TDMD, although capable of compensating the TUT-DIS3L2 machinery in controlling the steady state level of miRNAs, are apparently less robust. We therefore hypothesize that the TUT-DIS3L2 machinery may function in situations that require rapid changes in miRNA abundance. We are actively testing this hypothesis and hope to report the result in a future study.

REVIEWERS' COMMENTS:

Reviewer #1 (Remarks to the Author):

My concerns have been addressed.

Reviewer #2 (Remarks to the Author):

I have carefully read and assessed the response to all reviewers comments. The authors addressed the questions and comments well and I am happy to suggest this work for publication.

Reviewer #3 (Remarks to the Author):

The authors answered all my concerns and I have no further questions.